# Laissez-Faire Stallions? Males’ Fecal Cortisol Metabolite Concentrations Do Not Vary with Increased Female Turnover in Feral Horses (*Equus caballus*) [note 1]

**DOI:** 10.3390/ani13010176

**Published:** 2023-01-03

**Authors:** Maggie M. Jones, Cassandra M. V. Nuñez

**Affiliations:** 1Department of Natural Resource Ecology and Management, Iowa State University, 2310 Pammel Dr., Ames, IA 50011, USA; 2School of Natural Resources and Environment, University of Florida, P.O. Box 116455, Building 0724, 103 Black Hall, Gainesville, FL 32611, USA; 3Department of Biological Sciences, The University of Memphis, 3774 Walker Avenue, Ellington Hall, Room 239, Memphis, TN 38512, USA

**Keywords:** cortisol, *Equus caballus*, female turnover, immunocontraception, stress response

## Abstract

**Simple Summary:**

Stress can come in many forms for social animals. We investigated the effects of increased social instability on individual male stress levels in feral horses. When females change groups (thereby leaving one male to join another) stallions will often fight with one another and will remain alert, watching out for rival stallions and monitoring female locations; all potentially stressful behaviors. Such responses can be particularly important in managed populations as contracepted females are more likely to change groups, causing increased social instability in both the groups left and joined. We collected fecal samples from stallions to determine if decreased social instability (caused by females) affected male cortisol metabolite concentrations (an indicator of the physiological stress response). Surprisingly, we found that stallions experiencing increased social instability did not exhibit higher fecal cortisol even while they engaged in the fighting and alert behaviors typically associated with stress. Our results highlight the importance of considering both physiological and behavioral measures when investigating animal responses to challenging situations. Only by using such a holistic approach can we best understand the potential costs animals face; this is especially important in managed populations in which human perturbation can lead to unintended side effects.

**Abstract:**

Stress responses can be triggered by several physical and social factors, prompting physiological reactions including increases in glucocorticoid concentrations. In a population of feral horses (*Equus caballus*) on Shackleford Banks, North Carolina, females previously immunized with the immunocontraceptive agent porcine zona pellucida (PZP) change social groups (bands) more often than unimmunized females, disrupting the social stability within the population. We assessed the effects of increased female group changing behavior (or female turnover) on individual male stress by comparing fecal cortisol metabolite (FCM) concentrations among stallions experiencing varying amounts of female group changing behavior. FCM concentrations did not significantly correlate with female turnover. Similarly, FCM concentrations were not dependent upon the timing of female group changing behavior. These findings suggest that female turnover rate has little influence on physiological measures of stress in associated stallions. That said, Shackleford stallions experiencing increased female turnover do engage in behaviors typically associated with stress (increased vigilance, highly escalated male-male conflicts). Future work should compare FCM concentrations across time within populations and among populations managed under different strategies to better isolate factors influencing stallion stress physiology. Such studies are especially important if we are to determine how changes in female behavior related to immunocontraception impact physiological and behavioral indicators of stress for non-target animals. Finally, our study highlights the importance of considering both physiological and behavioral measures when investigating animal responses to potentially challenging situations.

## 1. Introduction

The mammalian stress response can be triggered by various social stimuli, from dominance contests to territorial intrusions, presenting a potential cost of group living [1]. Experiencing such stressors prompts a multi-faceted physiological response that includes an increase in glucocorticoids [2]. This is beneficial when experienced over a short period of time; for example, it can trigger behavioral responses that allow animals to avoid or escape potentially dangerous situations [2,3]. However, chronic stress can be detrimental to an animal’s health, possibly resulting in lower fertility, suppressed immune responses, and neurological defects [3,4,5,6] but see [7,8].

The effects of social challenges on physiological stress may be more pronounced in highly social species such as feral horses (*Equus caballus*). Feral horses live in groups, called bands, consisting of one or sometimes multiple males (or band stallions), one or more females (or mares), and their offspring (or foals) [9,10]. These bands are typically stable, with females staying with the same band stallion for most of their adult lives [11]. However, this social stability can be disrupted by changes in individual behavior within feral horse populations [12,13,14]. On Shackleford Banks, an island off the North Carolina coast, female behavior has changed in conjunction with contraception management with porcine zona pellucida (PZP), with immunized females changing bands up to ten times more often than unimmunized females [12,13]. Although active PZP treatment has been largely suspended on Shackleford, previously immunized females experiencing prolonged subfertility continue to demonstrate this increased group-changing behavior [15,16]. Females exhibit increased fecal cortisol metabolite (FCM) concentrations during group changes, and those making two or more group changes continue to show higher FCM concentrations up to two weeks after their last group change [17]. However, the potential effects of increased group-changing behavior on the physiology of these mares’ associates have not yet been studied. Here, we examine links between female group changing behavior (or female turnover) and the physiological stress responses of their associated band stallions (hereafter referred to as stallions).

Facets of the social environment including social instability, aggression, presence of estrous females, and population density have been linked to increased male glucocorticoid concentrations in several species [1,18,19]. In such species, social instability can arise during the mating season when new individuals immigrate into social groups or when dominance hierarchies are unstable. This instability often produces increased cortisol levels among males, likely due to increased competition for access to females [18,19,20]. For example, in gray-cheeked mangabeys (*Lophocebus albigena*), male-male aggression and the presence of females in estrus are correlated with elevated cortisol levels, and immigrant males exhibit higher levels than residents [18]. Similar periods of social instability are experienced by the feral horses on Shackleford Banks when previously immunized females immigrate into or emigrate from different bands at high rates [12,13]. Additionally, as females previously immunized with PZP are less likely to become pregnant, some continue cycling throughout the breeding season, extending male courting behavior, and potentially exacerbating and/or prolonging band instability [14,16,21]. These periods of instability are likely to trigger stress responses not only among group-changing females but also among members of affected bands, particularly the stallions.

Changes in the predictability of female movement with increased group-changing behavior can also result in increased intrasexual competition among stallions [22,23]; such activity has been documented in the Shackleford population [15,24]. Similar increases in male aggression are often associated with elevated cortisol levels in other species [25,26,27]. If stallion aggression increases in response to increased mare turnover, competing stallions may exhibit comparable increases in cortisol.

Such effects do not occur in isolation, however. For example, aspects of the physical environment, such as habitat visibility, can influence both individual behavior [10] and physiological responses [28]. In feral horses, habitat visibility has been shown to increase stallion mate-guarding behavior and territorial defense, likely because open habitats allow for greater visibility of rival stallions [10,15]. Additionally, in arctic ground squirrels (*Urocitellus parryii*), individuals in open habitats exhibited higher fecal cortisol levels than those in closed habitats, potentially due to an increase in perceived predation risk in high visibility areas [28]. Effects of the physical environment should therefore be considered when assessing animals’ stress responses.

We examined whether stallions experiencing group-changing behavior and/or exhibiting energetically costly behaviors to retain mares have higher stress levels and are in worse overall physical condition. To this end, we assessed the relationships between FCM concentrations, the number of female group changes experienced by stallions, stallion-stallion aggression, and body condition in the context of males’ surrounding environment. We hypothesized that stallions experiencing more group changes (both into and out of their groups) would have higher FCM concentrations and lower body condition scores than those experiencing fewer group changes. We also predicted that there would be a positive effect of stallion-stallion contest rate on FCM concentrations. Finally, we hypothesized that stallions in areas with high visibility, where they can more easily and consistently see rivals, would have higher FCM concentrations than those in low visibility areas. By assessing the effects of female turnover on stallion stress levels, we aim to provide a broader understanding of the behavioral and physiological consequences of decreased social stability in this free-living, highly social species.

## 2. Materials and Methods

### 2.1. Study Site and Subjects

The focal population of feral horses lives on Shackleford Banks, a barrier island off the coast of North Carolina, USA. The horses are habituated to human presence as Shackleford is visited by approximately 100,000 tourists annually [29]. The island is approximately 15 km long and ranges from 0.2 to 1.2 km wide. Several vegetation zones run across the length of the island with sandy beach bordering the ocean followed by dunes covered primarily with sea oats (*Uniola paniculata*), flat swales with patchily distributed grasses, maritime forest, and salt marsh dominated by cordgrass (*Spartina* spp.) [10] (personal observation). Historically, Shackleford has been defined by three distinct ecological regions. The East is primarily flat and open with a few small dunes and limited sources of freshwater [10] (personal observation). In the Middle, the terrain is also flat and generally open with small stands of maritime forest and more evenly distributed water sources [10] (personal observation). Finally, the West is dominated by high dunes. dense brush, and maritime forest and has two primary water sources [10] (personal observation). The study was conducted during two breeding seasons; due to travel logistics, observations were made from June through August in 2016 and from May through August in 2017. In both years, we observed all stallions and their females: 21 stallions and 70 females were organized into 18 bands, and 20 stallions and 69 females were organized into 19 bands in 2016 and 2017, respectively. Three stallions and their associated females were observed in 2016 only; two stallions and their females were observed in 2017 only; the remaining 18 stallions and their females were observed in both years. Six and 14 foals were born in 2016 and 2017, respectively. Data on foals are not presented in this paper. Most bands had one dominant stallion except for two double-stallion bands in 2016 and one double-stallion band in 2017. Individuals were identified with freeze brands, distinctive markings (e.g., facial markings), coloration, and other identifying features. All individuals were reliably distinguished with assistance from a catalog of individuals created by the National Park Service (NPS), and NPS personnel were available to assist in identification when necessary (Dr. Sue Stuska, personal communication).

### 2.2. PZP Contraception

This feral horse population is federally protected under the Shackleford Banks Wild Horses Protection Act of 1997 and is managed jointly by the National Park Service and the Foundation for Shackleford Horses [30]. To maintain the population at the predetermined size of approximately 120 animals, the National Park Service began managing the population via immunocontraception in 2000. Designated mares received initial and booster doses of ZonaStat-H, a PZP vaccine (Science and Conservation Center, Billings, MT, USA), which were administered annually from February through April (Dr. Sue Stuska, personal communication). Injections contained 100 µg of PZP plus an adjuvant, and initial doses contained Freund’s Complete Adjuvant, Modified *Mycobacterium butyricum* (Calbiochem #344 289, San Diego, CA, USA). Booster doses contained Freund’s Incomplete Adjuvant (Sigma #F5506). In 2010, when the NPS realized that some mares immunized repeatedly experienced prolonged subfertility, PZP treatment was suspended to allow the population to increase (Dr. Sue Stuska, personal communication). The NPS maintains detailed records concerning the timing of PZP treatment, which has been shared with the authors (Dr. Sue Stuska, personal communication).

### 2.3. Behavioral Data

Behavioral data and fecal samples were collected primarily by one observer (M.M.J. Jones) and were supplemented with additional data from six other observers (M. Fatka, R. Schwartzbeck, A. Jewel, M. Stuckenschneider, M. Stuckenschneider, and C.M.V. Nuñez). All observers were trained by C.M.V. Nuñez. All bands were located twice per week, and GPS location and the identity of all individuals present were noted. As witnessing group-changing behavior by females was rare (*n* = 1), the presence and absence of females in each band were noted to monitor this behavior. These data were used to calculate the number of group changes (the total number of times females entered or left a band) experienced by stallions.

In addition, we recorded instances of stallion-stallion aggression as they occurred (ad libitum) to ensure sufficient sample sizes for these rare behaviors [31]. Other forms of behavioral sampling would not have been sufficient: scan sampling would have precluded the recording of behaviors that occurred outside the scan (data collection) period, while focal sampling would have precluded the recording of data for more than one animal simultaneously, resulting in the potential omission of aggressive encounters by other individuals [31]. Stallion-stallion contests consisted of a range of behaviors, from visual assessments such as parallel prancing, to high-intensity physical contact such as biting and front and rear kicks [32]. Full descriptions of behaviors that we included as contest behaviors are described as follows (in order of escalation level based on a scale of one to six developed by Hynes [32]):Visual displays including parallel prance, pawing the ground, neck arch, and approach-retreat without physical contact.Olfactory assessment including nose-to-nose and nose-to-genital contact and fecal pile sniff without physical contact.Auditory assessment using aggressive vocalizations (squeals).Low-intensity physical contact including pushing and kick threats.Moderate-intensity physical contact including hind and front kicks that may result in injury.High-intensity physical contact including biting and rearing while kicking with the front legs that are likely to result in injury.

To ensure that the animals were not disturbed by our presence, all observations were conducted from at least 15–20 m away as advised by the NPS and observers remained still and silent during all sampling [33]. As the Shackleford horses are largely habituated to human presence, these distances are not disruptive. Upon our approach, the horses typically continued their activity or looked to the observer for some number of seconds before returning to their previous activity (personal observation). As behavioral activity can vary by time of day, we observed horses at all times of day from 6:00 am to 6:00 pm. However, due to logistic restrictions, most observations were concentrated in the middle of the day. The twenty-three stallions were observed for a total of 292 h over both years; each stallion was observed for a mean of 7.1 h per season ± 0.7 (range = 1.2–20.5).

### 2.4. Visibility

We quantitatively assessed visibility across the island as this factor may influence stallion behavior and interband conflict [10]. Visibility was quantified through viewshed analysis in ArcMap 10.4.1 (https://support.esri.com/en/products/desktop/arcgis-desktop/arcmap/10-4-1, accessed on 14 November 2018) [34,35]. Digital surface models were created using USGS LiDAR point cloud data (https://www.sciencebase.gov/catalog/item/5bba28ebe4b0fc368eabf5b9, accessed on 1 September 2018) to account for both bare earth elevation and vegetation structure [36]. We used Garmin eTrex 10 units to collect GPS locations at every band sighting from the observer’s location (as close to the horses’ elevation as possible). An observer offset of 1.2 m (the average height of Shackleford horses) was added to each observation point to account for the height at which a horse would be able to see across the landscape at eye level (Dr. Sue Stuska, personal communication). The Visibility tool in ArcMap was then used to predict the number of 10 m × 10 m cells visible from each point, and the number of visible cells was averaged across all points for each band and converted to km^2^ to produce an average visibility score for the band.

We analyzed this average visibility score rather than tying visibility to individual samples because it was more comparable to our measure of interest: female turnover experienced by stallions over the entire field season. Additionally, given their large home ranges and the variation in visibility across the island, horses can move between areas with different levels of visibility from day to day. Furthermore, FCMs reflect concentrations produced approximately 24 h prior to sample collection [37]; therefore, there was likely a mismatch between the “current” and “previous” habitat visibility experienced by stallions on the day of collection versus the day of FCM production, making an average visibility score for the entire home range a more appropriate metric.

### 2.5. Fecal Sampling

We collected fecal samples in 2016 and 2017 during our behavioral observations as in Nuñez et al. [17]. Samples were collected ad libitum [31] and only when we visually confirmed the animal’s identity and the sample’s location on the ground. Within minutes of defecation, we randomly selected and homogenized 2–3 fecal balls and stored them in 20 mL vials at a 2.5:1 ratio by volume of 95% ethanol to feces [38]. We recorded the time and date of collection for all samples. When possible, the homogenized feces were separated into 2 samples per fecal deposit. Samples were stored in a cooler at ambient temperature (from 1–5 days) until they were taken off the island and frozen at −80 °C. This storage technique has been proven a valid method of preserving fecal samples in the field and was used in a previous study of FCM concentrations in Shackleford mares [17,38]. Samples were shipped to Iowa State University on dry ice. We collected a total of 171 samples from stallions (85 samples from 17 stallions in 2016 and 86 samples from 18 stallions in 2017). Each stallion contributed 2.59 ± 0.36 (range = 1–6) and 2.61 ± 0.42 (range = 1–7) samples in 2016 and 2017, respectively.

### 2.6. Steroid Analysis

We extracted FCMs from samples using methods modified from Kozlowski et al. [39]. After the ethanol evaporated off the samples in a fume hood, we vortexed 0.5 g of fecal matter into 5.0 mL of a 50/50% methanol/extraction buffer solution. Preservation of fecal samples with ethanol and subsequent evaporation of the ethanol for fecal steroid analysis is a common procedure in field studies as the immediate freezing of samples is often not possible [38,40,41,42]. Extraction buffer contained 0.15M NaCl, 0.04M NaH_2_PO_4_•H_2_O, and 0.06M Na_2_HPO_4_ in deionized water, with 0.1% RIA grade bovine serum albumin and 0.1% sodium azide and was brought to pH 7.0 by addition of either NaOH or HCl. Extraction buffer was kept at 4 °C and allowed to come to room temperature before use. After vortexing, samples were shaken for approximately 16 h at 200 RPM. We let the fecal matter settle for 1 h, decanted the solution and centrifuged samples on a Beckman Coulter Allegra X30 centrifuge (Beckman Coulter, Brea, CA, USA) for 1 h at 4000 RPM, after which the supernatant was removed and stored at −80 °C until assay. To validate the assay for use in feral stallions, we homogenized a collection of ten samples taken from random stallions to make a sample pool. We extracted as outlined above and assayed the pool to determine parallelism with the EIA standard curve (see below).

We analyzed FCM concentrations using the Enzo Assays Cortisol ELISA kit (ADI-901-071, Enzo Life Sciences Inc., Farmingdale, NY 11735, USA), following the manufacturer’s instructions. We diluted all samples 1:13 in assay buffer and split each diluted sample into duplicate wells. We performed reruns for any duplicates with a coefficient of variation (CV) greater than 15% (%CV = (standard deviation/mean) × 100). We validated the assay as in Brown et al. [43]. A serial dilution of a stallion fecal pool showed parallelism to the cortisol standard curve (Linear Model; Cortisol concentration (pg/mL) × Sample type (standard or pool): F1,8 = 0.47, *p* = 0.51). Intra- and inter-assay CVs were 5.5 ± 0.5% (mean ± SE) and 5.0%, respectively (*n* = 7). Mean assay recovery ((observed/expected) × 100) was 102 ± 6.0%. All samples were analyzed in the same manner, allowing for comparison of relative FCM concentrations among Shackleford horses.

Although we were not able to validate our assay with an adrenocorticotrophic hormone (ACTH) challenge with these animals, we performed a biological validation of our methods with samples from lactating and non-lactating females (also collected during this study). Lactation is physically demanding [44] and lasts for 12–18 months (on average) for Shackleford mares (Nuñez, unpublished data), potentially leading to chronic stress and higher overall FCM concentrations [45,46]. When compared, lactating females did indeed exhibit higher FCM concentrations than did non-lactating females during this study (Linear Mixed Effects Model: estimate = 1.11; SE = 0.30; *t* = 3.73; *p* = 0.02), strongly supporting the effectiveness of our fecal collection, extraction, and assay protocols.

### 2.7. Body Condition

We visually assessed body condition through rump scoring, using a scale of one to five, similar to that described by Rudman and Keiper [47]. All stallions were scored an average of 2.3 ± 0.2 times in 2016 and 3.8 ± 0.3 times in 2017.

### 2.8. Statistical Analysis

We used general linear mixed effects models in R (version 3.4.1) to examine how female turnover influenced stallion FCM concentrations [48]. The FCM concentration of each sample (ng FCMs/g feces) was averaged across the two samples collected for each fecal deposit, resulting in a total sample size of 91. This average FCM concentration was used as the response variable for all models. The main independent variable of interest, the number of female group changes stallions experienced, was significantly correlated with the total time stallions were observed (Linear Model: estimate = 0.30; SE = 0.04; *t* = 6.28; *p* < 0.0001) (see Appendix A, Table A1 for the raw number of group changes and time observed for each stallion). We, therefore, used the residuals from that model as an independent variable when determining the effects of the number of group changes males experienced on their FCM concentrations. Models with raw group change data yielded similar results (see Appendix A).

Models of stallion FCM concentrations also included the timing of sample deposition regarding group changes. Sample timing was established by classifying each sample as collected before the stallion experienced a female group change (1–2 weeks before the change; *n* = 22), during a group change (within 1–2 weeks of the change; *n* = 51), or after the stallion experienced a group change (within 3–5 weeks after the change; *n* = 6). The timing for each sample was determined using the date of sample collection and the approximate dates of mare group changes. Group change date was estimated as the median date between the last day a female was observed with a stallion and the next day that female was seen with a different stallion.

We were also interested in the effect of contests between stallions on their FCM concentrations. As with female turnover (see above), residuals were calculated to control for the significant correlation between the frequency of stallion-stallion contests and the total time stallions were observed (Linear Model: estimate = 0.10; SE = 0.22; *t* = 4.75; *p* < 0.0001). Sample timing was established as noted above by classifying each sample as collected before (*n* = 31), during (*n* = 38), or after (*n* = 6) the stallion engaged in a contest with another stallion.

All models also included the following factors likely to affect stallion behavior and FCM concentrations as fixed effects: stallion age, which is correlated with social rank and physiological status; average female age (within bands), as younger females are more likely to switch groups; and the average number of adult females (aged 4+ years) present (there was low variability in the number of females within bands throughout each season (SE = 0.44 (0–1.09)), which is related to the amount of effort required for stallions to retain females within their bands [16,19,49,50,51]. Visibility was included as Shackleford stallions have demonstrated increased aggression towards mares in areas with higher visibility [15]. We also included the time of day that samples were collected (expressed with the hour as an integer from 0 to 24 and minutes as fractional hours (e.g., 3:30 pm = 15.5)) as FCM concentrations often follow diurnal patterns, with concentrations in domestic horses typically peaking in early morning, gradually decreasing throughout the day, with the lowest concentrations at night [52,53]. All models included stallion identity as a random effect to control for pseudo-replication and differences in baseline FCM concentrations between individual stallions. Finally, we also included year in the models to control for variation in behavior and steroid concentrations between the two years of the study. Covariates in the models were not strongly correlated (all *r* values < 0.40). A significance level of 0.05 was used to evaluate the effects of predictor variables on stallion FCM concentrations. To check the data for outliers, we used Cook’s distance measures, which identified two observations with potentially large influences on the model. However, removing these points from the dataset did not change the interpretation of model results.

While we also intended to use general linear mixed effects models to assess the effect of FCM concentrations on stallion body condition, there was very little variation in body condition scores within the population. Therefore, we could not perform the appropriate statistical tests. On a scale from one to five, stallions had an average body condition score of 3.7 ± 0.1 (range = 2.35–4.0) and 3.4 ± 0.2 (range = 1.3–4.0) in 2016 and 2017, respectively.

## 3. Results

A total of 148 female group changes were recorded throughout the study (36 group changes in 2016 and 112 in 2017). On average, stallions experienced 2.12 ± 0.51 (range = 0–6) and 6.22 ± 1.03 (range = 0–15) group changes in 2016 and 2017, respectively. Stallion FCM concentrations were not associated with residual group changes experienced (the number of group changes above or below that predicted from a linear model of group changes ~ times stallions were observed, see 2.8. Statistical analysis) (Figure 1; Linear Mixed Effects Model: estimate = 2.61, SE = 1.89, *t* = 1.40, *p* = 0.17). Similarly, FCM concentrations did not vary with female group change timing (before, during, or after a change) (Linear Mixed Effects Model: (during) estimate = −15.41, SE = 11.39, *t* = −1.35, *p* = 0.18; (after) estimate = 7.58, SE = 18.16, *t* = 0.42, *p* = 0.68). However, stallion FCM concentrations did vary with habitat visibility and with year: stallions exhibited higher concentrations in areas of lower visibility (Linear Mixed Effects Model: estimate = −21.57, SE = 10.18, *t* = −2.12, *p* = 0.04), and higher concentrations in 2017 compared to 2016 (Figure 2A; Linear Mixed Effects Model: estimate = 31.62, SE = 8.98, *t* = 3.52, *p* = 0.0008); the latter results coincided with an increase in female group changing behavior in 2017 (Figure 2B; Linear Mixed Effects Model: estimate = 1.12, SE = 0.32, *t* = 3.50, *p* = 0.0009). A model including raw data for the number of group changes males experienced yielded similar results (See Appendix A).

Sixty-eight stallion-stallion contests were observed over the course of the study (25 in 2016 and 43 in 2017); on average, stallions engaged in 1.47 ± 0.40 (range = 0–5) and 2.40 ± 0.31 (range = 0–5) fights per season in 2016 and 2017, respectively. Stallion FCM concentrations were not associated with the residual number of contests (the number of contests above or below that predicted from a linear model of stallion–stallion contests ~ times stallions were observed, see 2.8. Statistical analysis) (Linear Mixed Effects Model: estimate = 0.41, SE = 9.10, *t* = 0.04, *p* = 0.96). Similarly, FCM concentrations did not vary with contest timing (Linear Mixed Effects Model: (during) estimate = −20.73, SE = 19.90, *t* = −1.50, *p* = 0.14; (after) estimate = −5.17, SE = 19.46, *t* = −0.27, *p* = 0.80). A model including raw data for the number stallion–stallion contests yielded similar results (see Appendix A).

## 4. Discussion

Contrary to our predictions, the number of group changes males experienced did not significantly affect stallion FCM concentrations (Figure 1 and Figure A1). Though the number of group changes experienced by stallions (Figure 2B) and stallion FCM concentrations (Figure 2A) were both higher in the second year of the study, we cannot definitively determine the cause of this increase, as there are several social and environmental factors that could impact stallion stress levels in this system [12,13,15,54,55,56]. Similarly, FCM concentrations did not vary with the (1) timing of group changes by mares; (2) number of contests stallions engaged in; or (3) timing of stallion-stallion contests. These findings are somewhat surprising given that stallions experiencing more female group changes engage in more frequent and intense contests with other stallions and spend more time vigilant than those experiencing fewer group changes [15], behaviors commonly associated with increases in FCM concentrations in other species [25,27]. In addition, male-male conflict is associated with increased cortisol levels in several taxa including red deer (*Cervus elaphus*), golden hamsters (*Mesocricetus aurutus*), and rainbow trout (*Oncorhynchus mykiss*) [25,27,57], and vigilance behaviors are associated with increases in glucocorticoids in many species, including horses and other ungulates [58,59,60]. While our findings suggest that increased female turnover does not significantly influence stallion stress levels in the Shackleford population, increases in antagonistic and vigilance behaviors by stallions experiencing higher female turnover [15] could induce a stress response that is either not integrated physiologically or one that is obscured by confounding factors [54,55,58,61]. Conversely, the logistical limitations of our study may have affected our ability to detect a physiological response. For example, because FCM concentrations do not reflect short-term changes in glucocorticoids [37], they may not be the ideal metric with which to assess the effects of relatively short-lived behavioral responses to female turnover or intra-sexual competition. Additionally, the fact that field logistics limited our ability to precisely determine the dates that females changed groups may have also limited our ability to accurately detect the effects of this factor on male FCM concentrations.

Interestingly, males living in areas with lower overall visibility exhibited increased FCM concentrations. We had predicted the opposite based on our previous result showing that Shackleford stallions living in low visibility areas exhibit less aggression towards their females [15]. Upon further investigation, we found that stallions living in areas of lower visibility also (1) experience fewer group changes (Linear Mixed Effects Model: estimate = 2.59, SE = 0.77, *t* = 3.40, *p* = 0.002) and (2) engage in fewer contests with one another (Linear Mixed Effects Model: estimate = 0.97, SE = 0.38, *t* = 2.60, *p* = 0.01) than do stallions living in areas of high visibility. Given these additional behavioral associations, we again would have expected to see lower FCM concentrations in stallions living in low visibility areas. It is likely that stallions living in these areas are less able to closely monitor females and detect rivals at a distance [10]; this inability may induce higher baseline FCM concentrations, but see [28]. However, these results remain surprising given the fact that these animals experience lower levels of female turnover and fight with one another less frequently than do their counterparts living in high visibility areas.

Taken together, our results highlight the importance of considering multiple metrics—including behavioral, physiological, and ecological measures—if we are to comprehensively assess animal welfare in the wild. The use of conventional glucocorticoid concentration measurements through plasma, saliva, or fecal samples to assess animal fitness and wellbeing is not straightforward. The relationship between FCM concentrations and individual or population fitness is often highly context-dependent [62]. Moreover, the ultimate impact of changes in FCM concentrations on animal welfare depends on the actions of the steroids within specific tissues and the physiological and behavioral consequences of those actions [63]. There are myriad ways in which cortisol can act upon physiology and behavior, and the effects of increases in cortisol depend on the specific stressor and environment as well as the animal’s developmental stage, sex, and species [63,64,65]. Moreover, FCM concentrations are not the sole indicator of stress, and when used as such can be misleading [58]. In some instances, increases in stress-indicative behaviors are observed without an associated increase in glucocorticoids [58,61]; in others, they are associated with a decrease in glucocorticoids [66]. Therefore, behavioral indicators, such as the antagonistic and vigilance behaviors discussed above, should be used in conjunction with physiological measures to provide a more comprehensive assessment of animal stress level [67,68,69]. Moreover, habitat features are rarely considered when assessing animal welfare; our results indicate the importance of assessing habitat features as they may affect animal responses in unexpected ways. Future research including field studies of natural stressors’ effects on FCM concentrations, such as this one, should be combined with laboratory studies concerning the impact of circulating cortisol on target tissues and neurological pathways. Studies comparing these effects on individuals of different sexes and developmental stages living in varying habitats will improve our understanding of the potential consequences of elevated FCM concentrations for animal health and welfare [63].

## 5. Conclusions

While our findings do not indicate a correlation between variation in stallion FCM concentrations and social instability, stallions experiencing increased female turnover engage in more highly escalated stallion-stallion contests and spend more time vigilant [15], behaviors associated with increased cortisol in several species [25,27,57,58,59,60]. Given this mismatch between physiological and behavioral indicators of stress in this and other animal populations [58,61,66], we emphasize the importance of evaluating multiple measures of stress to provide a clearer, more complete picture of the consequences that social instability and other potentially stressful conditions can have on animal populations. Moreover, habitat features can influence animal responses in counterintuitive ways; as such, it is critical that their potential effects are assessed. Such considerations are especially important for managed populations as the maintenance of animal welfare is often commensurate with more effective management practice [16,70,71].

## Figures and Tables

**Figure 1 animals-13-00176-f001:**
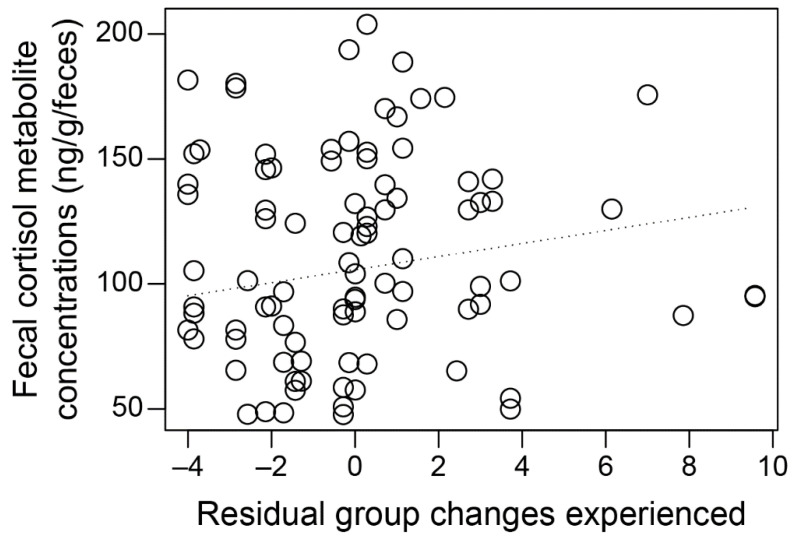
Residual group changes stallions experienced and stallion FCM concentrations. FCM concentrations did not vary with the number of group changes males experienced. Open circles indicate individual fecal samples; dotted line indicates model predictions.

**Figure 2 animals-13-00176-f002:**
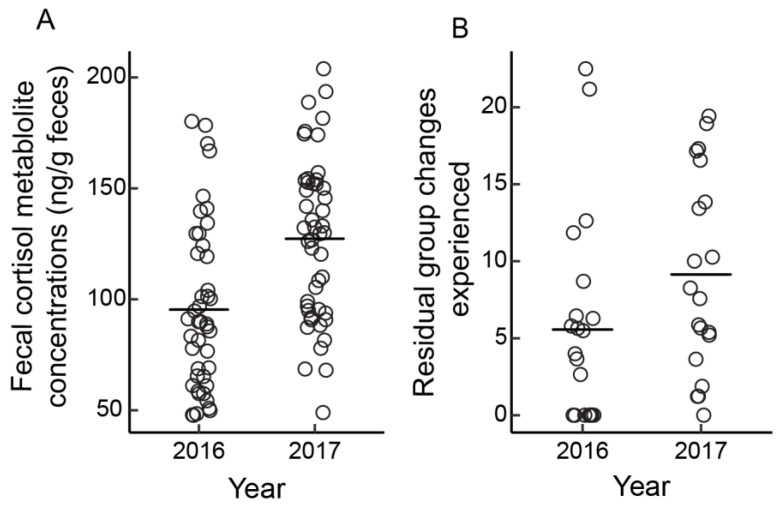
Differences in (**A**) FCM concentrations and (**B**) the number of group changes stallions experienced in the two years of the study. FCM concentrations and the number of group changes were both higher in 2017 than in 2016. Points are jittered to allow clear visualization of all (**A**) fecal samples and (**B**) individual stallions.

## Data Availability

The data presented in this study are openly available in the Open Science Framework repository at https://osf.io/dthk5/ (accessed on 1 September 2018) and will be made public upon the paper’s acceptance.

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
