# Peer review of "Laissez-Faire Stallions? Males’ Fecal Cortisol Metabolite Concentrations Do Not Vary with Increased Female Turnover in Feral Horses (Equus caballus)†"

_animals, 2023, doi:10.3390/ani13010176_

Round 1

Reviewer 1 Report

This paper describes how FCM concentrations in feral horse stallions are affected by factors such as females leaving their group and fights between stallions. Although high group change rates were found in this population, the only significant factor was year. Cortisol levels were higher in the year with more group changes. As FCM are increasingly used to examine stress levels in a wide variety of species, this paper highlighting the importance of collecting behavioral observations to complement fecal samples is important.

This paper is well written, and clearly lays out the rationale behind the study, the hypotheses and objectives, and the results. The methods were lacking in some details. Three phrases were used: group changes, group change rates, and female turnover. These were not defined and although they were used interchangeably it was not clear whether they were the same thing. It was not clear whether group changes experienced by a male involved females leaving his group, joining his group, or both. See also my comment on L354-355. If I understand correctly, on average 7 mares moved between stallions every day during the breeding season. This seems incredibly high, and makes me wonder if you can even call a group a band with that level of instability. Group changes are central to the paper and it is critically important that it is made crystal clear to the reader exactly what is being measured.

I would like to see results with group change rate as a continuous variable, or blocked into smaller bins, especially to compare males with 0 group changes with those with higher group change rates. This could be added to the appendix if it does not provide more detail, but I think it is important to include. Your lack of significant results could simply be because you were looking at too broad a scale. Did you consider transforming data in Fig A1 and A2 to adjust the distribution?

Your results could also be explained by the time period surrounding a change being too broad to capture that event from background chronic stress. You did not describe how you dealt with situations when groups changes occurred multiple times – what happened if a change happened weekly?

In the analyses it is important to state in the methods whether correlations among variables were assessed.

L15 – change “their” to “one” or “a” male.

L16 – make “location” plural.

L25 – holistic does not start with w.

L76 – add “associated” before “band stallions”.

L141 – rephrase to something like “Data on foals are not presented in this paper”. Presumably you did observe foals (as in you saw them), any data collected are just not relevant to this manuscript.

L167 – clarify that study bands equate to all bands present, or else qualify how many.

L182 – remove the second “are” at the beginning of the line.

L202 – state how many individual stallions total. It would be useful to define a band stallion earlier, and make it clear that henceforth these band stallions are just referred to as stallions. This will mean the reader is not confused that some may be bachelors or subadults.

L238 – be consistent whether you call them band stallions, stallions, or males. Presumably the males discussed here are the stallions discussed on L202? Change in terminology makes this unclear.

L239 – rephrase to “Each male contributed…” for clarity.

L326 – clarify whether classification followed the same time breaks as for female group changes.

L330 – define an adult female.

L354-355 – I don’t quite understand how this math works. If there were 31 group changes at a rate of 7 per day, then that would mean the horses were observed for ~4 days. It would be useful to have a definition of a group change, and to make clear to the reader whether this is the same as female turnover. I assume a group change to be mare A moves from stallion 1 to stallion 2. But she may also be accompanied by mare B. Would this count as two group changes (mare A + mare B) or one (mare A and B together)? Was this included as a group change for stallion 1 (losing mares) or stallion 2 (gaining mares) or both? Were any stallions left without any mares at all?

L385 – references here are not in the journal format.

L435 – remove “our”.

L443 – presumably “Please add:” will be removed in copy editing.

Table A1 – in the text it says rate was calculated by days observed, whereas this table shows hours observed. It would be more useful if the table showed data in the same order of magnitude as used in data analyses. It would also be useful to show over how many days these hours of observation were collected.

Author Response

Please see attached file with our responses.

Thank you for our assistance in this process!

Reviewer 2 Report

The manuscript presents observations of stallions behavior in the closed/island population of feral horses during two reproduction seasons 2016 and 2017. As a stress factor fecal level of cortisol was measured. The modified method of cortisol extraction was described.

1.The title, abstract and the aim of the study are clear and logical.

2.Materials and methods are correct

3.The statistical analyses are described in details

4.presentation of the results-it should improved

5. Discussion and citations are correct.

However, lack of the clear correlation between behavior and fecal concentration of cortisol make the manuscript less interesting, also Authors are not able to explain such result.

My suggestions are:

1. calculate the results separate for 2016 and 2017 (in this latter time of the observations was almost double comparing to 2016 year-why?)

2.try to compare the RIA and ELISA methods for cortisol measurement

3.the time of collecting samples is very important - beginning or end of reproduction season 

In summary, the results should be calculated again with consideration of additional factors.

Author Response

Please see our attached responses. 

Thank you for your help in the publication process!

Round 2

Reviewer 2 Report

Dear Authors,

Thank you for your answers which are reasonable and adequate to my suggestions.

Yours sincerely,

Reviewer